# Multidimensional Machine Learning Model to Calculate a COVID-19 Vulnerability Index

**DOI:** 10.3390/jpm13071141

**Published:** 2023-07-15

**Authors:** Paula Andrea Rosero Perez, Juan Sebastián Realpe Gonzalez, Ricardo Salazar-Cabrera, David Restrepo, Diego M. López, Bernd Blobel

**Affiliations:** 1Research Group in Telematics Engineering, Telematics Department, Universidad del Cauca, Popayán 190002, Colombia; parosero@unicauca.edu.co (P.A.R.P.); jsrealpe@unicauca.edu.co (J.S.R.G.); ricardosalazarc@unicauca.edu.co (R.S.-C.); dsrestrepo@unicauca.edu.co (D.R.); dmlopez@unicauca.ed.co (D.M.L.); 2Medical Faculty, University of Regensburg, 93053 Regensburg, Germany; 3eHealth Competence Center Bavaria, Deggendorf Institute of Technology, 94469 Deggendorf, Germany; 4First Medical Faculty, Charles University Prague, 12800 Prague, Czech Republic

**Keywords:** COVID-19, dataset, machine learning, vulnerability index

## Abstract

In Colombia, the first case of COVID-19 was confirmed on 6 March 2020. On 13 March 2023, Colombia registered 6,360,780 confirmed positive cases of COVID-19, representing 12.18% of the total population. The National Administrative Department of Statistics (DANE) in Colombia published in 2020 a COVID-19 vulnerability index, which estimates the vulnerability (per city block) of being infected with COVID-19. Unfortunately, DANE did not consider multiple factors that could increase the risk of COVID-19 (in addition to demographic and health), such as environmental and mobility data (found in the related literature). The proposed multidimensional index considers variables of different types (unemployment rate, gross domestic product, citizens’ mobility, vaccination data, and climatological and spatial information) in which the incidence of COVID-19 is calculated and compared with the incidence of the COVID-19 vulnerability index provided by DANE. The collection, data preparation, modeling, and evaluation phases of the Cross-Industry Standard Process for Data Mining methodology (CRISP-DM) were considered for constructing the index. The multidimensional index was evaluated using multiple machine learning models to calculate the incidence of COVID-19 cases in the main cities of Colombia. The results showed that the best-performing model to predict the incidence of COVID-19 in Colombia is the Extra Trees Regressor algorithm, obtaining an R-squared of 0.829. This work is the first step toward a multidimensional analysis of COVID-19 risk factors, which has the potential to support decision making in public health programs. The results are also relevant for calculating vulnerability indexes for other viral diseases, such as dengue.

## 1. Introduction

COVID-19 is a disease caused by a virus called SARS-CoV-2, which was first reported on December 31, 2019, upon warning of a cluster of viral pneumonia cases in Wuhan, China [1]. The first case was confirmed in Colombia on 6 March 2020 [2]. The COVID-19 confirmed testing rates provide critical information to understand the full impact of the pandemic and identify ways to reduce morbidity and mortality. 

The National Administrative Department of Statistics (DANE) is responsible for planning, collecting, processing, analyzing, and disseminating official statistics in Colombia. In this context, DANE published in 2020 an index of vulnerability to COVID-19 called the “Index of Vulnerability by Block”, using the country demographic variables and comorbidities. The data used were solely obtained from the National Population and Housing Census 2018 (CNPV) and the discharge summaries from the Individual Health Services Register [3]. Climatological, environmental, socioeconomic, and mobility factors, among many others, are variables reported in the literature that were not considered when creating the vulnerability index by DANE. In what follows, the impact of the aforementioned factors will be discussed in more detail.

Several studies have analyzed the impact of climatological and environmental parameters, considering the geographical location of each country and intending to demonstrate the possible relationship between environmental factors and morbidity and mortality due to COVID-19 on the other side [4]. At the global level, there is great evidence on this subject; however, at the national level, there is very little information. In the same direction, other studies have shown that vaccines against COVID-19 effectively protect against the comorbidities associated with this disease, including mortality [5]. Since immunization began worldwide, many countries have performed evaluations of the rates of hospitalization and death from COVID-19 among vaccinated and unvaccinated persons, intending to calculate the effectiveness of vaccination schemes [6]. In this sense, vaccination data can also be useful for determining a vulnerability index. However, DANE did not consider this information, which may be related to the fact that there were not enough vaccination data in Colombia when they calculated their vulnerability index. 

Furthermore, the loss of income from work due to unemployment caused by COVID-19 resulted in an increase in poverty rates and income inequality among people in vulnerable conditions, such as informal workers, women and indigenous youth, afro-descendants, and people with disabilities. People in conditions of socioeconomic vulnerability are at greater risk of infection and death from COVID-19 since inequalities are directly related to their ability to protect themselves from infection. A higher incidence of comorbidities is also associated with greater severity of the disease and even death. The percentage of the overcrowded population also plays an important role because areas with a higher proportion of overcrowded people are more affected by COVID-19 [7]. The inclusion of the aforementioned impact factors in the calculation of a new vulnerability index will lead to different results and interpretations, influencing its level of certainty.

Considering the context mentioned above, the following research question arose: how can a COVID-19 vulnerability index for Colombia be determined which considers COVID-19 case data published daily by the National Institute of Health, and other relevant risk factors in addition to those proposed by DANE? 

The general objective of this research was to propose a machine-learning (ML) model to calculate a COVID-19 vulnerability index that considers human, environmental, sociodemographic, and socioeconomic risk factors as well as the database of historical cases of COVID-19 in an integrated manner. The outcome can assist decision making in public health programs. For that purpose, we developed a base ML model with information similar to that used by DANE (except for the information on comorbidities, which was not publicly available due to privacy concerns). The aim was to compare how close the base vulnerability index was to the reference vulnerability index obtained by DANE. Then, a multidimensional index, including several risk factors that were not included in the DANE index, was proposed to support the decision-making process of health agencies and make it possible to identify vulnerability to COVID-19 in the country’s main cities. The additional data sources taken into account additionally to those of the DANE vulnerability index were COVID-19 data including vaccination data from the Ministry of Health as the main source, unemployment and gross domestic product information from DANE, and mobility, climatological, and spatial information from satellite images. This manuscript is an extended version of the paper published in the IOS SHTI pHealth 2022 Proceedings called “Risk Factors for COVID-19: A Systematic Mapping Study” [8].

The subsequent sections of the document are organized as follows. Section 2 presents the materials and methods used in this research. Section 3 presents the results of the research. Section 4 discusses the results obtained. Finally, Section 5 presents the conclusions and future work. 

## 2. Materials and Methods

This section includes 4 sections: the identification of relevant COVID-19 risk factors; the development of the base model; the development of the multidimensional index; and the evaluation of the proposed multidimensional index.

### 2.1. Identification of Relevant Risk Factors of COVID-19

A systematic mapping was performed to identify the risk factors of COVID-19, followed by an analysis of the results.

To identify the risk factors, a search for review articles was performed using the Scopus database, which has broad coverage of scientific research, where 1786 related studies were identified and reviewed, of which 564 met the inclusion criteria. As the documents were analyzed, similar characteristics were detected among them, such as the type of review and the factors they covered, so two classifications were generated in the mapping. The first classification was called “type of research” and referred to the type of review conducted. For example, some reviews conducted experimental studies called “Review and experimentation”. Some reviews did not conduct studies but fulfilled the objective of a review called “Review”. Finally, some reviews did not indicate or follow a systematic review methodology in the abstract. Therefore, they were classified as “Non-formal reviews”.

The second classification was called “research context” and referred to the types of factors found and the possible combinations among them. This classification included the following categories: Human risk factors. They refer to people’s health conditions.Sociodemographic and socioeconomic risk factors. These indicate the characteristics of the population.Environmental factors. These factors include environmental variables.Sociodemographic and socioeconomic risk factors, and human factors.Sociodemographic and socioeconomic risk factors, and environmental risk factors.Sociodemographic and socioeconomic risk factors, environmental, and human risk factors.

Subsequently, an analysis of the results was performed, in which the risk factors were highlighted. The map of studies obtained in the systematic mapping, presented in Figure 1, allows identifying that the largest number of documents corresponds to the “review” type and just addresses human risk factors. The second largest number of documents, also corresponding to the “review” type, includes studies in the category of sociodemographic, socioeconomic, and human risk factors. The “review and experimental” group is the least dominant on the map. Concerning environmental risk factors, they are the least represented in the documents reviewed. In addition, among all the “type of research” categories, very few results were obtained for papers addressing environmental and human risk factors. The “non-formal review” type presents low dominance in the systematic mapping, and in the research context, human risk factors were the most prevalent ones for this category. Finally, it should be noted that the risk factors with the greatest presence in the documents were comorbidities such as diabetes, hypertension, obesity, and cardiovascular disease, but also age and sex.

Continuing with identifying risk factors, research was conducted on vulnerability indexes developed in the country (Colombia) and internationally to find out what had been done in other studies and the variables that had been considered. 

Regarding the vulnerability indexes for COVID-19, the most relevant were the following:C19VI vulnerability index [9]. This index was developed in the United States by the Center for Disease Control and Prevention (CDC), considering the following variables to calculate the index: socioeconomic status, household composition, disability, minority status and language, type of housing and transportation, and epidemiological and health system factors.As a result, a vulnerability map was obtained in which each city was identified with a color according to the level of vulnerability found.Vulnerability index of Colombia (DANE) [3]. DANE provided a COVID-19 vulnerability index with a geographic disaggregation level by blocks. The study’s objective was to categorize which people, according to the block where they live, have a higher probability of complications in case of infection by COVID-19. For this purpose, demographic characteristics and health conditions were considered. The variables used to calculate the DANE vulnerability index are presented below:
oComorbidities: hypertension, diabetes, ischemic heart disease, chronic pulmonary disease, and cancer.oDemographic characteristics: identification of people over 60, households in overcrowded rooms and bedrooms, and households at high and medium intergenerational risk per block.

Based on these variables, a series of steps were performed to consolidate a database of 407,277 rows with the columns above. After that, the K-means cluster analysis was applied, allowing the blocks to be grouped according to demographic characteristics and comorbidities.

Finally, the result obtained was the vulnerability map of Colombia, which shows the vulnerability to COVID-19 by block.

Considering the systematic mapping and the search of existing vulnerability indexes, it was found that the index developed by the CDC was the most appropriate reference model to this work. In this sense, a vulnerability index was constructed, called the “Base model”, in which variables of the CDC vulnerability index were considered, except those regarding comorbidities, because open access to these data is restricted in Colombia due to confidentiality concerns. The CNPV 2018 datasets, also developed by DANE, were the main data source to construct the index.

### 2.2. Base Model

The stages suggested by the Cross-Industry Standard Process for Data Mining (CRISP-DM) methodology were followed for constructing the base model. The CRISP-DM stages are business understanding, data understanding, data preparation, modeling, evaluation, and deployment (this last stage was not performed) [10]. 

First, a dataset was built in which 5 tables of the CNPV 2018 were used (housing, households, deceased, persons, and georeferenced data [11]). The objective of the base dataset was to determine whether it was necessary or not to create a new dataset that considers other types of risk factors and, therefore, to propose a new vulnerability index. These data were in the comma-separated values (CSV) format and classified by department (regional geographical units in which Colombia is divided). There are 32 departments in the country. The variables taken into account were the following: type of housing, number of bedrooms per household, number of deceased per household, the total number of persons in the household, sex of the dead (male or female), age of the deceased, sex of the person, age of the person in five-year groups, ethnic recognition, speaking the native language of his/her people, speaking other native languages, quality of health service provision, literacy, highest educational level achieved, and economic activity performed during the past week (variable focused on asking if the person worked and received income from work). The dataset also includes a variable called “COVID-19 vulnerability,” which contains the COVID-19 vulnerability data published by the DANE index aggregated by each municipality in the country (1104 municipalities and 19 special districts). This is a discrete value from 0 to 5, where 0 means no vulnerability, and 5 means high vulnerability. This variable was used as a dependent variable in the ML model evaluation phase to calculate the vulnerability values already calculated by DANE.

The dataset was processed using the Python Pandas library. The data was cleaned and pre-processed to eliminate variables that did not contribute to the objective of the work. The municipalities that did not include a value for vulnerability (output variable) were eliminated, thus resulting in a dataset of 89 columns and 1103 rows. This dataset can be found on Kaggle’s web platform as the “Base COVID-19 Dataset” [12]. 

In the modeling phase of the base index development process, confusion matrices were generated to observe the correlation between the dependent variable (output variable) and the independent variables. For this purpose, Pearson’s and Spearman’s correlations were considered. Pearson’s correlation evaluates the linear relationship between two quantitative variables [13]. Pearson’s coefficient indicates the variables’ association, and its value can take values between −1 and 1. Therefore, no linear relationship exists when variables show a correlation around zero. Spearman’s correlation was applied, because there was a monotonic relationship between the dependent and independent variables [14]. Nevertheless, the correlation between the variables was still close to zero. 

Classification algorithms were considered for the CRISP-DM evaluation phase because they allow the prediction of discrete or qualitative outputs [15]. The objective of the base model was to evaluate whether the obtained dataset performs an efficient vulnerability prediction by a municipality. Therefore, classification algorithms were used because the vulnerability values, calculated by the DANE, are discrete variables. The aim was to determine how well these values could be predicted with the created model. The evaluation used two scenarios (dividing the dataset into different percentages for training and testing in each scenario). In scenario 1, 80% of the dataset was taken as training and 20% for testing. In scenario 2, 70% of the dataset was taken as training and 30% for testing. Considering that no correlation was perceived in the confusion matrices, six different models were used for evaluation. 

The first model implemented was a linear discriminant analysis (LDA), a supervised classification method where a predictive model is built to determine the group to which it belongs. The second model was a quadratic discriminant analysis (QDA). This model is used when the set of predictor variables to be classified has two or more classes. It is considered the equivalent of non-linear discriminant analysis. The third model was k-nearest neighbors (KNN), where a learning classifier algorithm can be used as a regression or classification algorithm. The fourth model was the decision tree classifier, a supervised learning algorithm mainly used in classification problems. The fifth model was Gaussian Naive Bayes (GNB), a probabilistic ML algorithm typically used as a classifier. Finally, the sixth model used a support vector machine (SVM) algorithm, which can be used for classification and regression. To evaluate the performance of the algorithms, the following metrics were used: F1 (average macro), precision, recall, and accuracy. 

The F1 metric considers the number of false positives and false negatives, calculating a weighted average between precision and sensitivity, thus obtaining a single score representing the two variables. The precision metric measures the accuracy of the classifier when predicting positive cases. It is calculated as the ratio between correct predictions and the expected number of correct predictions. The recall metric detects positive instances, also known as sensitivity. It is calculated as the ratio between correct and total positive predictions. The accuracy metric determines the classification accuracy, i.e., the ratio between correct predictions and total predictions [16].

The results of the evaluation of the base dataset are presented in Section 3. 

### 2.3. Multidimensional Index

The multidimensional index was constructed by adding new data from other types of variables to the dataset created for the base model. Thus, the multidimensional index considers sociodemographic, socioeconomic, environmental, and human factors. 

The first five stages of the CRISP-DM methodology were also followed for this index (as was done for the base model).

Figure 2 presents a flowchart to facilitate understanding of the construction process of the multidimensional index.

The data collection for the construction of this index used open data. The sources found made it possible to unify a target dataset with the following datasets: Gross domestic product (GDP). GDP is the standard value-added of producing a country’s goods and services during a period [17]. This dataset provides a broader visibility of what each region (department) contributes to the country yearly. Considering the relevant period of the COVID-19 pandemic, data for 2020 and 2021 were considered. This research used GDP at constant prices with an annual periodicity.Climatological data. This type of data has been identified as a factor that increases the risk of COVID-19. For this reason, temperature and precipitation data were sought through Google Earth Engine. This platform allows access to these data for all the principal municipalities in Colombia [18] daily.Vaccination percentage. For the vaccination data, the information provided by the Ministry of Health was taken into account, which presents a report made in Power BI [19] in which the vaccination percentage curve by the municipality can be visualized.Unemployment rate. This dataset was collected from the information published by the Great Integrated Household Survey (named GEIH) conducted by DANE. The survey information is presented for the capitals of 24 departments (out of 32 and one special district) and published quarterly [20].Mobility data. This dataset was collected from reports published by Google. These reports allow tracing movement trends over time in different categories of places: grocery and pharmacy, parks, transit stations, retail and recreation, workplace, and residential, taking as a reference the mobility of the 5 weeks between January 3 and February 6, 2020. The increase or decrease percentages of mobility-specific areas were calculated [21].COVID-19 vulnerability. Same as the previous index, it contains the COVID-19 vulnerability data published by DANE. Including these data allows having a representative value of the variables already measured with the national index, therefore reflecting data on comorbidities, information on older adults in households, and overcrowding places data.COVID-19 case data. This is the output variable, considering that the evaluation of the model was performed using multiple machine learning models to predict the incidence of COVID-19 cases in the main cities of Colombia. Data were obtained from information published by the Ministry of Health, where daily data reported in each of the municipalities of Colombia can be found in the CSV format [22].

Considering the temporal and geographical limitations of some of the data sources, it was determined that a quarterly data periodicity was most appropriate to work for the following cities: Quibdó, Cali, Cúcuta, Armenia, Popayán, Ibagué, Neiva, Florencia, Valledupar, Tunja, Riohacha, Bogotá, Villavicencio, Pereira, Manizales, Medellín, Santa Marta, Sincelejo, Montería, Pasto, Bucaramanga, Barranquilla, Cartagena, and San Andrés. Subsequently, the datasets were cleaned and pre-processed. It was necessary to transform the data into CSV files for the GDP, vaccination, and unemployment rates. Regarding the climatological data, an average was made to find each quarter’s temperature and precipitation values. Next, the mobility dataset was cleaned, deleting unnecessary data. Then, an average mobility dataset per municipality was made for each quarter. The complete and integrated dataset can be found on Kaggle’s web platform as the “Multidimensional index of COVID-19 Colombia” [23]. 

In the multidimensional index development modeling phase, confusion matrices were generated to determine the correlation between the independent and dependent variables (output variables). The results section (Section 3) shows the confusion matrices obtained.

Regarding the evaluation of this model, several supervised learning regression algorithms were applied because the output variable (incidence) is continuous [24]. The algorithms used were linear regression, decision trees, KNN, SVM, random forest, and gradient boosting. Two meta-estimators called Extra Trees Regressor and AdaBoost Regressor were also used. 

The same two scenarios (presented in the base model) were used to divide the dataset into training and tests. As the output variable was the COVID-19 incidence, the algorithm with the best performance in predicting this variable was evaluated. 

The multidimensional model is expected to perform better than the base model in estimating COVID-19 vulnerability. The root means square error (RMSE) and R-squared were used to assess the algorithm’s performance. The RMSE metric indicates how close the observed data points are to the predicted values. It can also be interpreted as the standard deviation of the unexplained variance. A low RMSE value indicates a better fit. R-squared suggests the fitness of the model. This metric takes values between 0 and 1, where 0 represents that the proposed model does not improve the prediction over the mean model, and 1 indicates perfect prediction [25]. Negative R-squared values are likely to occur; this situation arises in cases where the model is less fitted than the average [26].

To improve the performance of the multidimensional index models, it was decided to optimize the hyper-parameters of the algorithms with the best results in the first tests. The hyper-parameters’ optimization is presented in Appendix A.

### 2.4. Evaluation of the Multidimensional Index for Predicting the Incidence of COVID-19 

To complement the evaluation of the performance of the multidimensional index, the best-performing algorithms were used to predict the incidence of COVID-19 cases in the main cities of Colombia. The results are compared with the results of a reference model trained with the DANE vulnerability index (COVID-19 vulnerability column of the multidimensional dataset) as the independent variable and the COVID-19 incidence (COVID-19 case data column of the multidimensional dataset) as the dependent variable. This reference model is called a “Reference predictor”. The same algorithms were used to predict the cases in the experiment for the two models (multidimensional and reference predictor), i.e., linear regression, decision trees, k-nearest neighbor (KNN), support vector machine (SVM), random forest, and gradient boosting, as well as the two meta-estimators (Extra Trees Regressor and AdaBoost Regressor). Furthermore, the evaluation was performed with the same scenarios (training and evaluation percentages).

Section 3 presents the results of evaluating the multidimensional index for predicting the incidence of COVID-19.

## 3. Results

Confusion matrices are analyzed for the used variables in the multidimensional index. Later, the performance evaluation results of the base model and multidimensional indexes are presented. 

### 3.1. Confusion Matrixes for the Multidimensional Index

In the confusion matrix in Figure 3, the green color indicates a high positive correlation between variables, the fuchsia color indicates a high negative correlation, and the white color indicates a low correlation. For example, Figure 3 shows that variables such as GDP and “residential” (mobility) are positively correlated, while temperature and “retail_and_recreation” are negatively correlated. In addition, it is important to mention that the incidence column is the output variable. Figure 4 presents the confusion matrix for Spearman’s correlation.

In Figure 4 (similar to Figure 3) the last column shows some shades of green and some fuchsia, indicating these variables have a positive correlation (green) or negative (fuchsia). The high intensity of the green color shows a high correlation. In Figure 4, variables such as GDP, “residential”, “grocery and pharmacy”, and vaccination percentage have a positive correlation with the incidence variable. At the same time, the variables temperature and “workplaces” (mobility) have a negative correlation. In addition, variables such as precipitation and percentage of unemployment show a fairly low correlation with incidence.

### 3.2. Base Model Evaluation

For the base model, two scenarios were used: splitting the data into 80% training and 20% testing for the first scenario, and 70% training and 30% testing for the second scenario. However, the performance obtained for this model in predicting the reference DANE vulnerability index (COVID-19 vulnerability variable) was deficient for both scenarios. Therefore, in Table 1, only the scenario with the best performance is reported. 

Table 1 shows that the DecisionTreeClassifier algorithm performs best for the F1 metric. This value was obtained when the dataset was divided into 70% training and 30% test. It can also be noted that the base vulnerability index correctly predicts the vulnerability value by 87.3% among the total predictions. However, the precision and recall values are low, around 43%, indicating that the number of correct predictions compared to all predictions is very low. Because of this, the F1 value in the best performance model is 42.9%.

These results may be due to the lack of comorbidities in the base dataset.

### 3.3. Multidimensional Index Evaluation

The specific variables used in this multidimensional index were the following:Gross domestic product (GDP);Temperature;Precipitation;Vaccination percentage;Unemployment rate;Mobility in grocery and pharmacy;Mobility in parks;Mobility in transit stations;Mobility in retail and recreation;Mobility in workplace;Mobility in residential;COVID-19 vulnerability;COVID-19 case data as the output variable.

Table 2 shows the performance for the first scenario (80% training and 20% test) in the multidimensional index for predicting the COVID-19 vulnerability variable of the multidimensional dataset. Table 3 shows the model’s performance for this index’s second scenario (70% training and 30% test). Table 4 shows the results for the optimized models in this index. The procedure that was followed for the optimization of each model is presented in Appendix A.

In Table 2 and Table 3, it is possible to observe that the Extra Trees Regressor algorithm presents the best results in each of them, because it has an RMSE level close to 0 and an R-squared value closer to 1. It should be noted that in Table 3 (70% training and 30% test) the RMSE value closest to 0 is that of the Random Forest Regressor algorithm (0.008, while the RMSE value of the Extra Trees Regressor algorithm is 0.009), but the difference in the R-squared value between these two algorithms means that the best option, combined with the two values, is the Extra Trees Regressor algorithm. The best values (RMSE and R-squared) between the two proposed scenarios were obtained for the first scenario of 80% training and 20% test; therefore, this was used in the optimization of hyper-parameters.

In Table 4, in which the results of the algorithms with optimized hyper-parameters are presented, again the Extra Trees Regressor algorithm presents the best results in the optimized values of RMSE and R-squared (0.007 and 0.00829, respectively) using the first scenario (80% training and 20% test).

The extra trees are an extension of the random forest regression model and were proposed by Geurts et al. [27]. The extra trees belong to the class of decision tree-based ensemble learning methods. In decision tree-based ensemble methods, multiple decision trees are used to perform classification and regression tasks. The extra trees are less susceptible to overfitting and report better performance [27].

### 3.4. Results of the Multidimensional Index for Predicting Incidence of COVID-19

Table 5 shows the performance for the first scenario (80% training and 20% test) of the reference predictor used to estimate the real incidence of COVID-19 cases. Table 6 presents the performance for the second scenario (70% training and 30% test) in the DANE index. 

Table 7 presents a comparison between the results obtained in the different models used in the optimized multidimensional index (Table 4) and the results obtained for these same models in the DANE index (reference predictor) use the scenario with the better results 80%–20% (Table 5). It is possible to appreciate that the results obtained in the multidimensional index have better values for the two used metrics (RMSE and R-squared). It should be noted that the values of Table 2 and Table 5 are not compared because the models in Table 2 are not optimized, while in Table 4 they correspond to the optimized values explained in Appendix A.

In Table 2, Table 3, Table 5 and Table 6, some negative values are presented in the R-squared metric. Although it is normal for the values of this metric to be between 0 and 1, this situation can occur in cases where the model is less fitted than the average (as mentioned in [26]).

Next, the models with the best performance (for the multidimensional predictor and reference predictor) were used to predict the incidence of real COVID-19 cases reported by quarters. It is clarified that, when dividing the dataset into training and test, in each execution different data were selected, the common cities were selected in the execution of each model to compare the prediction. Figure 5 and Figure 6 show the results obtained. Figure 7 shows that the values predicted by the multidimensional index are closer to the expected values than those predicted by the DANE index.

## 4. Discussion

The presented research evaluated the performance of a multidimensional machine-learning model to calculate a COVID-19 vulnerability index that considers human, environmental, sociodemographic, and socioeconomic risk factors in an integrated manner. The vulnerability indexes creation required the evaluation of several ML algorithms to identify which one behaved better in predicting the incidence of COVID. This represents an important differentiation concerning other previous works in the literature in which results of only one algorithm for calculating the index are presented. Another representative difference of this work, concerning the consulted previous works, is the diversity of types of variables used because other works mainly considered human factors.

This research began by creating a base machine learning model with information similar to that used by DANE (except for the information on comorbidities, which was not publicly available due to privacy concerns). The aim was to compare how close the base vulnerability index was to the reference vulnerability obtained by DANE. Regarding the evaluation of the base model, it is important to mention that most of the algorithms used presented a high precision. However, the value of the F1 score metric was very low. This is because the F1 score is the average between the precision and recall metrics. Therefore, correct predictions relative to all predictions were low. In addition, although several sociodemographic and socioeconomic variables were considered for the base model, optimal results were not obtained. This could be because human risk factors, i.e., comorbidities (for which no data were available), which seem quite relevant, were not taken into account.

Therefore, the construction of a multidimensional index of COVID-19 was necessary. The new index added different types of risk factors to improve the performance of the vulnerability index. The proposed multidimensional index, without optimizing and after performing the hyper-parameter optimization, presents a high performance in predicting COVID-19 incidence. After optimization (a process that did not generate significant results), the R-squared and RMSE metrics with the multidimensional index obtained a maximum value of 0.829 and 0.007, respectively, with the Extra Trees Regressor algorithm. This algorithm is a meta-estimator, which calculates the best predictions from various decision trees. 

The Decision Tree Regressor algorithm was the one that most improved the value of the R-squared metric after adjusting the hyper-parameters, obtaining a tree in which the number of nodes and leaves was considerably reduced. This algorithm is relevant because, in the regression algorithms, the decision trees present a better behavior for the datasets used in this study.

Through optimization, it was possible to show that the internal CV performed by the GridSearchCV function caused the R-squared metric of the optimized algorithms not to exceed the metric value before optimization. Consequently, it was necessary to perform the evaluations manually to avoid the internal CV of the GridSearchCV function but considering in the evaluation the values of the hyper-parameters that the function returned.

In the case of the multidimensional index, it must be taken into account that although it may have better performance due to the variables added that provide a better context to the model, having a dataset with more characteristics can cause the results to be affected by overfitting. Having a model and dataset that span more dimensions and, therefore, have more variables also require having more samples for training to prevent the model from being more prone to memorizing patterns and, thus, reducing its ability to generalize. In this case, it is suggested to carefully select the most important variables for the model and pre-process them properly in order to have a dataset with the most relevant variables, eliminate possible noise, and reduce the risk of overfitting.

Testing the developed model in different countries, not only in Colombia as was done in the research, may be an interesting option to validate the model created. However, the acquisition of data of all the variables proposed in the model could be a costly process. In addition, it is very likely that the data for some variables cannot be acquired in another country with the same periodicity, or the necessary source is not available. Some data proposed in the model come from national sources, which cannot be obtained in other countries in the region or globally. Although the application of the model in other countries is difficult, exactly as it is proposed, it could be adapted depending on the available sources and the viable variables to use. 

In addition, this study used as a reference the index already created in Colombia by DANE in 2020 to compare the performance of the multidimensional index created to predict real COVID-19 cases in the country. The results showed that the metrics obtained by the multidimensional index were better than those of the reference predictor trained with the DANE vulnerability index. The above establishes that the multidimensional index performs a better prediction of the incidence of COVID-19 cases in the country. 

For the development of this research, three types of risk factors were determined: human risk factors, sociodemographic and socioeconomic risk factors, and environmental risk factors. Taking the above into account, the classification of the datasets used in the multidimensional index is presented below:Vulnerability variable. The variable of this dataset is human and sociodemographic type because it includes vulnerability data already calculated from comorbidities and characteristics of people such as age and overcrowding in homes.GDP. The variables of this dataset are sociodemographic and socioeconomic risk factors.Percentage of unemployment. The variables of this dataset are sociodemographic and socioeconomic risk factors.Vaccination percentage. The variables of this dataset are human risk factors.Mobility data. The variables of this dataset are sociodemographic and socioeconomic risk factors.Temperature and precipitation. The variables of this dataset are environmental risk factors.

Some limitations should be considered in this study, such as the fact that it was not possible to include the risk factor of comorbidities, which has been used in most of the vulnerability indexes found and has also been characterized in the literature as a risk factor. It is important to mention that due to data policies in Colombia it was not possible to add this variable in the base model. For the multidimensional index, this risk factor is indirectly included in COVID-19 vulnerability index. In addition, it is also worth noting that the multidimensional index does not consider data for 2022, and only handles data for 2020 and 2021. Furthermore, during the data collection phase of the base model, it became evident that access to the data was too limited to calculate an index that could consider various types of risk factors. In addition, the pre-processing and data cleaning required considerable work to implement the ML models. Despite the work done, the lack of access to data on variables resulted in the algorithms obtaining low values for the selected metrics. 

It is important to clarify that the proposed model only considered the main cities of Colombia, due to data acquisition limitations that exist for certain variables in cities with a low population in the country. The periodicity of most of the variables considered in the multidimensional index was quarterly; this can generate a certain level of error in the predicted data, considering that a shorter period of time would be ideal.

On the other hand, it should be noted that, in Colombia, no study has considered an index comprising variables that consider different types of risk. Additionally, at the international level, the vast majority of studies do not take into account environmental variables. This study showed that considering variables that belong to different types of risk factors generates an efficient prediction of the incidence of COVID-19. Likewise, the temporality of the data plays a key factor and has not been considered in most of the developed indexes, which handle a static temporality. 

In this study, different machine learning algorithms were used to predict the incidence of COVID-19 to know which one performs better according to the dispersion of the data, as opposed to most studies where mathematical methods have been applied or a single machine learning algorithm has been implemented.

Finally, although this study has developed a multidimensional index to predict the incidence of COVID-19 in Colombia, it should be taken into account that the characteristics of the data of the variables in Colombia are not the same as in other countries, so it would imply that the data of each country should be taken into account. However, this study is a valuable contribution to the organizations or entities in Colombia because it can assist in health decision making, predicting the value of the incidence with high values for the metrics. 

## 5. Conclusions

This study shows how using open data sources allows the construction of a multidimensional dataset (a dataset that integrates risk factors of different types). This approach generates a great effort in the pre-processing stage. There may also be a lack of data for some variables selected for different periods. Nevertheless, it is an interesting solution that made it possible to identify and evaluate other factors found in the literature as factors of relevance for calculating a COVID-19 vulnerability index. 

This research also included evaluating several ML models implemented with various algorithms. This evaluation was optimized, trying to improve the results. The whole process showed that the model using the Extra Trees Regressor algorithm was the best for predicting the incidence of COVID-19 in 24 Colombian department capitals. 

Finally, this research shows the need for comprehensive access to data in Colombia and the protocols for open and anonymized data. This way, different groups or organizations could collect information for research purposes. In addition, delays in data publication and data release in inconsistent formats such as the portable document format (PDF) are common. These problems were addressed in the development of this research by performing manual imputation regarding some variables. 

This work is a first step for future research, in which a greater number of variables considered as risk factors for COVID-19 could be regarded in the search for an index that increasingly performs better predictions of the incidence of COVID-19. It is also relevant for calculating vulnerability indexes for other viral diseases, such as dengue. 

For future work, other machine learning models could be evaluated, and the index could be extended to other countries. As subsequent work, the adaptation of the model proposed for COVID-19 in other countries is also recommended, considering the review of the acquisition of previously necessary variables in this new context. Another option for future work is related to the application of the proposed model in the same context (Colombia) in a different timeline, which is recommended to be performed in approximately 2 years. Finally, another interesting option for future work is to adjust the model for other types of diseases such as dengue, which would require a greater amount of resources than the previous options since it is necessary to evaluate whether the selected variables are adequate, or if some need to be replaced or removed.

In summary, this work exemplifies the healthcare transformation towards a multidisciplinary, knowledge-based, context-aware ecosystem [28].

## Figures and Tables

**Figure 1 jpm-13-01141-f001:**
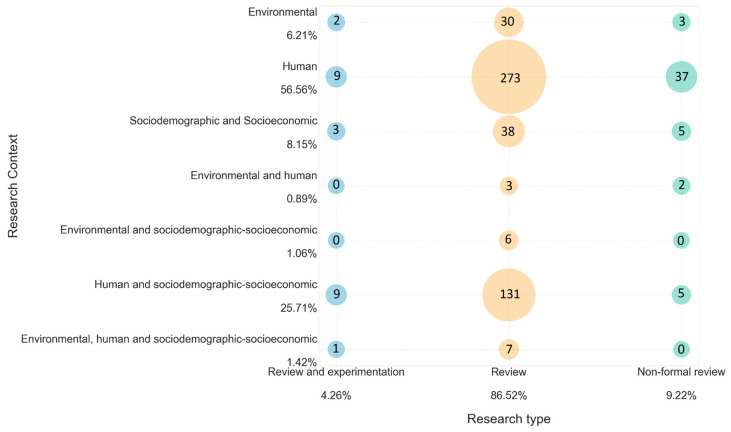
Bubble chart mapping and associating the type of research with the research context. Percentages are calculated for each axis.

**Figure 2 jpm-13-01141-f002:**
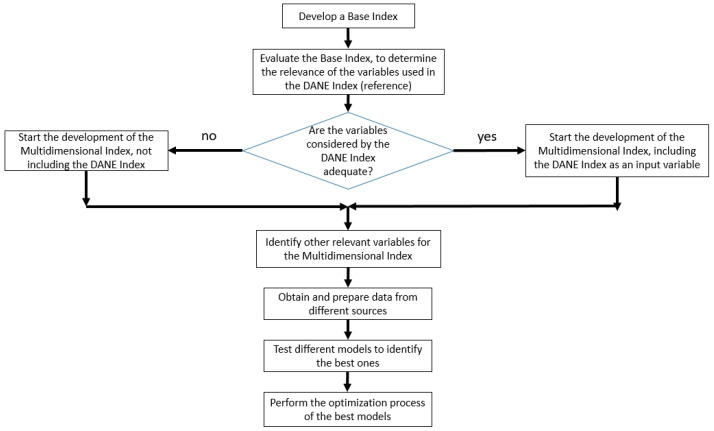
Flowchart of the multidimensional index construction.

**Figure 3 jpm-13-01141-f003:**
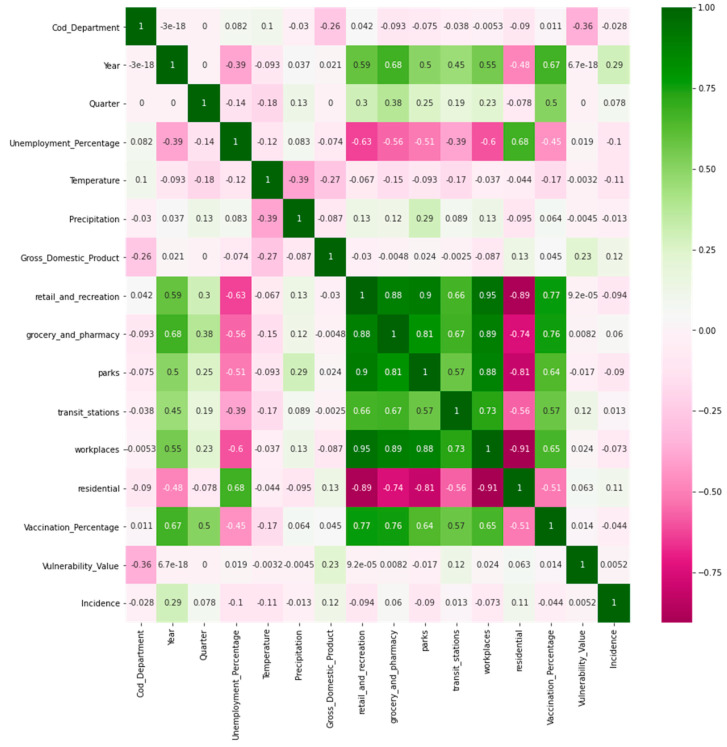
Confusion matrix (Pearson correlation).

**Figure 4 jpm-13-01141-f004:**
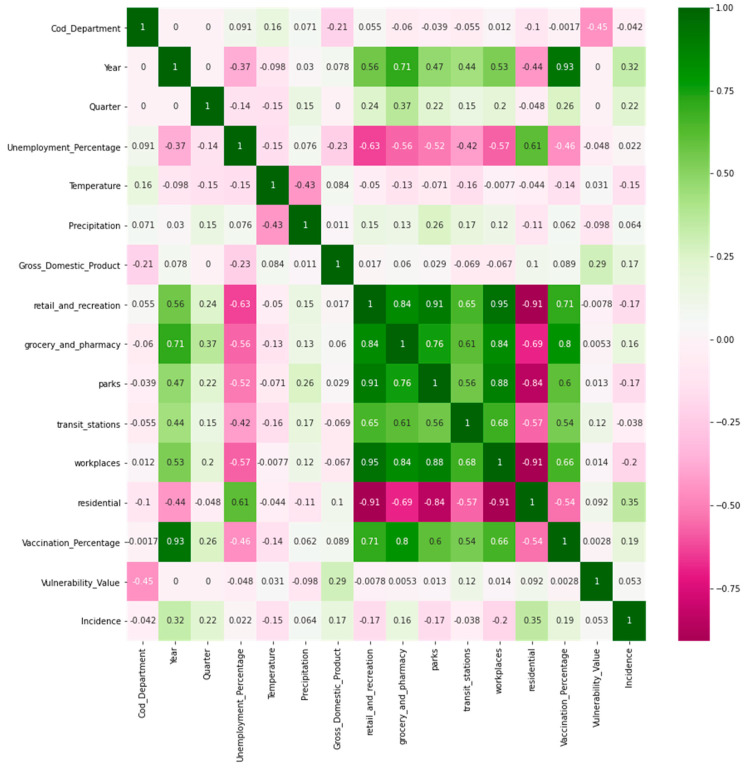
Confusion matrix (Spearman correlation).

**Figure 5 jpm-13-01141-f005:**
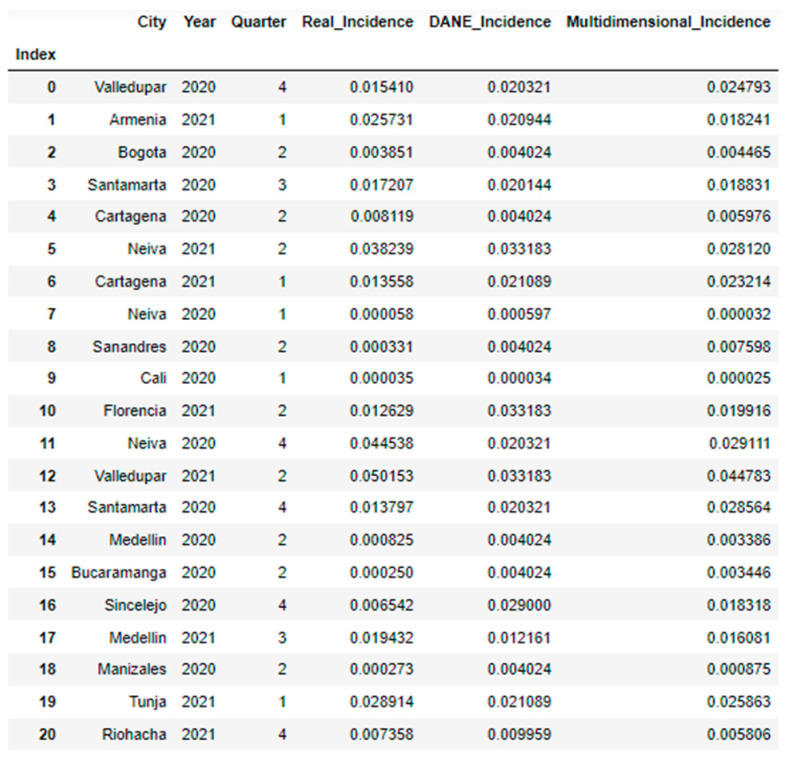
Table predicted vs. expected values, part 1.

**Figure 6 jpm-13-01141-f006:**
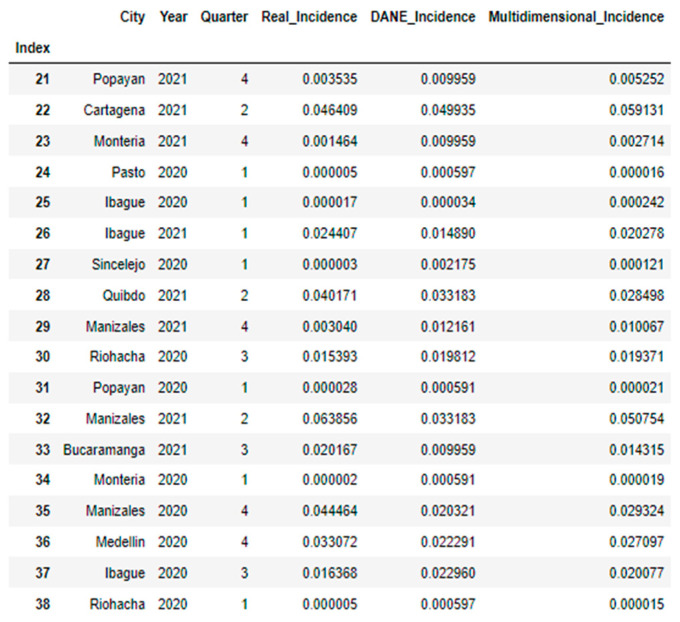
Table predicted vs. expected values, part 2.

**Figure 7 jpm-13-01141-f007:**
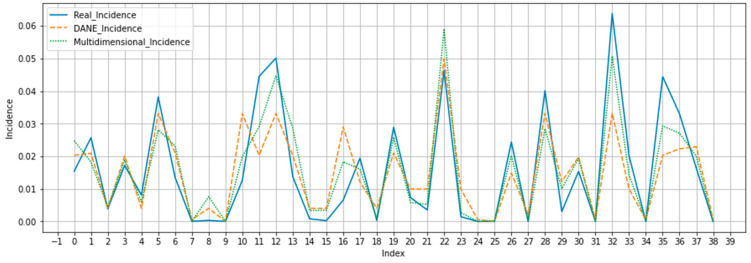
Graph of predicted versus expected values.

**Table 1 jpm-13-01141-t001:** 70% training and 30% test scenario (base model).

Model	F1 Score	Precision	Recall	Accuracy
LinearDiscriminantAnalysis	0.239	0.226	0.259	0.897
QuadraticDiscriminantAnalysis	0.192	0.185	0.2	0.927
KNeighborsClassifier	0.192	0.185	0.2	0.927
DecisionTreeClassifier	0.429	0.427	0.432	0.873
GaussianNaiveBayes	0.141	0.231	0.504	0.184
SupportVectorMachine	0.192	0.185	0.2	0.927

**Table 2 jpm-13-01141-t002:** 80% training and 20% test (multidimensional index).

Model	RMSE	R-Squared
Linear Regression	0.013	0.358
Decision Tree Regressor	0.010	0.611
K-Nearest Neighbor	0.019	−0.207
Support Vector Machine	0.033	−2.697
Random Forest Regressor	0.007	0.790
Gradient Boosting Regressor	0.008	0.758
Extra Trees Regressor	0.007	0.828
AdaBoost Regressor	0.008	0.761

**Table 3 jpm-13-01141-t003:** 70% training and 30% test (multidimensional index).

Model	RMSE	R-Squared
Linear Regression	0.013	0.319
Decision Tree Regressor	0.012	0.469
K-Nearest Neighbor	0.019	−0.335
Support Vector Machine	0.030	−2.496
Random Forest Regressor	0.008	0.720
Gradient Boosting Regressor	0.009	0.637
Extra Trees Regressor	0.009	0.700
AdaBoost Regressor	0.009	0.683

**Table 4 jpm-13-01141-t004:** Obtained results in the multidimensional index optimized, using 80% training and 20% test.

Model	Hyper-Parameters	Base R-Squared	Base RMSE	R-Squared of the Optimized Model	RMSE of the Optimized Model
Decision Tree Regressor	criterion = ‘absolute_error’ max_depth = 4 max_features = ‘auto’ random_state = 329 ccp_alpha = 7.179 × 10^−6^	0.611	0.010	0.708	0.009
Random Forest Regressor	max_depth = 13 max_features = 7 n_estimators = 125 random_state = 329	0.790	0.007	0.802	0.007
Gradient Boosting Regressor	learning_rate = 0.5 max_depth = 2 max_features = ‘auto’ n_estimators = 1000 n_iter_no_change = 5 random_state = 329 subsample = 1	0.758	0.008	0.765	0.008
Hist Gradient Regressor	learning_rate = 0.5 max_depth = 3	Does not apply	Does not apply	0.810	0.007
Extra Trees Regressor	n_estimators = 97 max_features = None random_state = 329	0.828	0.007	0.829	0.007
AdaBoost Regressor *	n_estimators = 247	0.761	0.008	0.811	0.007

* Model works internally with Decision Tree Regressor.

**Table 5 jpm-13-01141-t005:** 80% training and 20% test (reference predictor).

Model	RMSE	R-Squared
Linear Regression	0.016	0.090
Decision Tree Regressor	0.012	0.517
K-Nearest Neighbor	0.019	−0.287
Support Vector Machine	0.033	−2.697
Random Forest Regressor	0.010	0.608
Gradient Boosting Regressor	0.011	0.546
Extra Trees Regressor	0.011	0.561
AdaBoost Regressor	0.013	0.395

**Table 6 jpm-13-01141-t006:** 70% training and 30% test (reference predictor).

Model	RMSE	R-Squared
Linear Regression	0.016	0.033
Decision Tree Regressor	0.011	0.474
K-Nearest Neighbor	0.018	−0.220
Support Vector Machine	0.030	−2.496
Random Forest Regressor	0.010	0.596
Gradient Boosting Regressor	0.011	0.480
Extra Trees Regressor	0.011	0.535
AdaBoost Regressor	0.010	0.610

**Table 7 jpm-13-01141-t007:** Comparison between the models of the reference predictor and multidimensional index (using the best scenario, 80% training and 20% test).

Model	RMSE of Reference Predictor	RMSE of Multidimensional Index	R-Squared of Multidimensional Index	R-Squared of Reference Predictor
Decision Tree Regressor	0.012	0.009	0.708	0.517
Random Forest Regressor	0.010	0.007	0.802	0.608
Gradient Boosting Regressor	0.011	0.008	0.765	0.546
Extra Trees Regressor	0.011	0.007	0.829	0.561
AdaBoost Regressor	0.013	0.007	0.811	0.395

## Data Availability

The data presented in this study are available on request from the corresponding author.

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
