# Peer review of "Multidimensional Machine Learning Model to Calculate a COVID-19 Vulnerability Index"

_jpm, 2023, doi:10.3390/jpm13071141_

Round 1

Reviewer 1 Report

In the manuscript entitled “Multidimensional machine learning model to calculate a COVID-19 vulnerability index”, authors have developed a mathematical model based on ExtraTrees Regressor algorithm for prediction of vulnerability indexes of  disease using COVID-19 cases.

Reviewer thinks that model should be tested on data from at least two different places/ cities/ countries. Generally, any new model is supposed to be tested on different data sets for validation.

It is also suggested that authors can check the data available for some other disease and run the model for results. Data obtained from web-resources in different timeline can also be cross check point. Authors are mentioning this point is the conclusion stating that “calculating vulnerability indexes for other viral diseases, such as dengue”.

Authors should also write about the different applications of this mathematical along with the limitations and amount of possible errors.

Reviewer 2 Report

The purpose of this research conducted in Colombia was to create a comprehensive index for evaluating the risk and vulnerability associated with COVID-19. The National Administrative Department of Statistics (DANE) had already published a COVID-19 vulnerability index. However, the authors suggest listing the specific variables used in the multidimensional index , it would be beneficial to provide a concise list of these variables for clarity.

The methodology employed in the study incorporated various statistical techniques, resulting in a complex analysis with numerous tables and figures. To enhance readability, some tables could be relocated to the annex. The presence of an R command for Figures 2-7 appears unusual and should be rewritten  if necessary.

Although the ExtraTreesRegressor algorithm was identified as the top-performing model, the rationale for this selection remains unexplained. The applicability of these methods for calculating vulnerability indexes pertaining to other infectious diseases remains uncertain. Nevertheless, the study holds potential for informing decision-making processes within public health programs.

Reviewer 3 Report

The author extend the base vulnerability index to Multidimensional Index for Covid by identifying risk factors using meta analysis.  

Major concerns

1.      There should be a flow chart to describe how the Multidimensional Index was constructed, for readers not familiar with it.

2.      Line 326-331: “R-squared were used to assess the algorithm's performance which takes values between 0 and 1”

Is R-squared defined as one in linear regression? If so, then why there is negative (-0.22, -2.496) in Tbl 2-6. 

3.      The authors used the term "Optimized R-squared" in Tab 4 without defining it. Is that related the "R-sq of the optimized model/algorithm"? But that's very different meaning. Is the R-sq adjusted for the no. of parameters or not?

4.      “Optimized model” was mentioned many times, however, in what sense was not clearly defined. To me, the selection of splitting data by 80% or 70% for training has nothing to do with optimization.

5.      Researchers normally pick either 80-20% or 70-30% training-testing splitting, and then compared the performance among methods. I saw the comparison among LDF/KNN et al., however, missing the direct comparison between the base (vulnerability) index to the multidimensional index in any table or figures. Is the results from Tbl 2 (multidimensional) and Tbl 5 (base) comparable? If so, why not merge them?

6.      Although the warning overfitting for decision tree had been mentioned in the text, the potential overfitting from base index to multidimensional index can be further addressed in Discussion.   

Minors

7.      It’s unorthodox to have python coding (Fig 2-7) in the main text. Should be moved to appendix

Round 2

Reviewer 3 Report

The authors had satisfactory and fully replied the comments item by item.